# Counterfactual Fairness

**Matt Kusner** *
The Alan Turing Institute and
University of Warwick
mkusner@turing.ac.uk

**Joshua Loftus** *
New York University
loftus@nyu.edu

**Chris Russell** *
The Alan Turing Institute and
University of Surrey
crussell@turing.ac.uk

**Ricardo Silva**
The Alan Turing Institute and
University College London
ricardo@stats.ucl.ac.uk

## Abstract

Machine learning can impact people with legal or ethical consequences when it is used to automate decisions in areas such as insurance, lending, hiring, and predictive policing. In many of these scenarios, previous decisions have been made that are unfairly biased against certain subpopulations, for example those of a particular race, gender, or sexual orientation. Since this past data may be biased, machine learning predictors must account for this to avoid perpetuating or creating discriminatory practices. In this paper, we develop a framework for modeling fairness using tools from causal inference. Our definition of *counterfactual fairness* captures the intuition that a decision is fair towards an individual if it the same in (a) the actual world and (b) a counterfactual world where the individual belonged to a different demographic group. We demonstrate our framework on a real-world problem of fair prediction of success in law school.

## 1 Contribution

Machine learning has spread to fields as diverse as credit scoring [20], crime prediction [5], and loan assessment [25]. Decisions in these areas may have ethical or legal implications, so it is necessary for the modeler to think beyond the objective of maximizing prediction accuracy and consider the societal impact of their work. For many of these applications, it is crucial to ask if the predictions of a model are *fair*. Training data can contain unfairness for reasons having to do with historical prejudices or other factors outside an individual's control. In 2016, the Obama administration released a report[2] which urged data scientists to analyze "how technologies can deliberately or inadvertently perpetuate, exacerbate, or mask discrimination."

There has been much recent interest in designing algorithms that make fair predictions [4, 6, 10, 12, 14, 16–19, 22, 24, 36–39]. In large part, the literature has focused on formalizing fairness into quantitative definitions and using them to solve a discrimination problem in a certain dataset. Unfortunately, for a practitioner, law-maker, judge, or anyone else who is interested in implementing algorithms that control for discrimination, it can be difficult to decide *which* definition of fairness to choose for the task at hand. Indeed, we demonstrate that depending on the relationship between a protected attribute and the data, certain definitions of fairness can actually *increase discrimination*.

In this paper, we introduce the first explicitly causal approach to address fairness. Specifically, we leverage the causal framework of Pearl [30] to model the relationship between protected attributes and data. We describe how techniques from causal inference can be effective tools for designing fair algorithms and argue, as in DeDeo [9], that it is essential to properly address causality in fairness. In perhaps the most closely related prior work, Johnson et al. [15] make similar arguments but from a non-causal perspective. An alternative use of causal modeling in the context of fairness is introduced independently by [21].

In Section 2, we provide a summary of basic concepts in fairness and causal modeling. In Section 3, we provide the formal definition of *counterfactual fairness*, which enforces that a distribution over possible predictions for an individual should remain unchanged in a world where an individual's protected attributes had been different in a causal sense. In Section 4, we describe an algorithm to implement this definition, while distinguishing it from existing approaches. In Section 5, we illustrate the algorithm with a case of fair assessment of law school success.

## 2 Background

This section provides a basic account of two separate areas of research in machine learning, which are formally unified in this paper. We suggest Berk et al. [1] and Pearl et al. [29] as references. Throughout this paper, we will use the following notation. Let $A$ denote the set of *protected attributes* of an individual, variables that must not be discriminated against in a formal sense defined differently by each notion of fairness discussed. The decision of whether an attribute is protected or not is taken as a primitive in any given problem, regardless of the definition of fairness adopted. Moreover, let $X$ denote the other observable attributes of any particular individual, $U$ the set of relevant latent attributes which are not observed, and let $Y$ denote the outcome to be predicted, which itself might be contaminated with historical biases. Finally, $\hat{Y}$ is the *predictor*, a random variable that depends on $A, X$ and $U$, and which is produced by a machine learning algorithm as a prediction of $Y$.

### 2.1 Fairness

There has been much recent work on fair algorithms. These include fairness through unawareness [12], individual fairness [10, 16, 24, 38], demographic parity/disparate impact [36], and equality of opportunity [14, 37]. For simplicity we often assume $A$ is encoded as a binary attribute, but this can be generalized.

**Definition 1** (Fairness Through Unawareness (FTU)). *An algorithm is fair so long as any protected attributes $A$ are not explicitly used in the decision-making process.*

Any mapping $\hat{Y} : X \to Y$ that excludes $A$ satisfies this. Initially proposed as a baseline, the approach has found favor recently with more general approaches such as Grgic-Hlaca et al. [12]. Despite its compelling simplicity, FTU has a clear shortcoming as elements of $X$ can contain discriminatory information analogous to $A$ that may not be obvious at first. The need for expert knowledge in assessing the relationship between $A$ and $X$ was highlighted in the work on individual fairness:

**Definition 2** (Individual Fairness (IF)). *An algorithm is fair if it gives similar predictions to similar individuals. Formally, given a metric $d(\cdot, \cdot)$, if individuals $i$ and $j$ are similar under this metric (i.e., $d(i, j)$ is small) then their predictions should be similar: $\hat{Y}(X^{(i)}, A^{(i)}) \approx \hat{Y}(X^{(j)}, A^{(j)})$.*

As described in [10], the metric $d(\cdot, \cdot)$ must be carefully chosen, requiring an understanding of the domain at hand beyond black-box statistical modeling. This can also be contrasted against population level criteria such as

**Definition 3** (Demographic Parity (DP)). *A predictor $\hat{Y}$ satisfies demographic parity if $P(\hat{Y}|A = 0) = P(\hat{Y}|A = 1)$.*

**Definition 4** (Equality of Opportunity (EO)). *A predictor $\hat{Y}$ satisfies equality of opportunity if $P(\hat{Y} = 1|A = 0, Y = 1) = P(\hat{Y} = 1|A = 1, Y = 1)$.*

These criteria can be incompatible in general, as discussed in [1, 7, 22]. Following the motivation of IF and [15], we propose that knowledge about relationships between all attributes should be taken into consideration, even if strong assumptions are necessary. Moreover, it is not immediately clear

for any of these approaches in which ways historical biases can be tackled. We approach such issues from an explicit causal modeling perspective.

## 2.2 Causal Models and Counterfactuals

We follow Pearl [28], and define a causal model as a triple $(U, V, F)$ of sets such that

- $U$ is a set of latent **background** variables, which are factors not caused by any variable in the set $V$ of **observable** variables;
- $F$ is a set of functions $\{f_1, \ldots, f_n\}$, one for each $V_i \in V$, such that $V_i = f_i(pa_i, U_{pa_i})$, $pa_i \subseteq V \setminus \{V_i\}$ and $U_{pa_i} \subseteq U$. Such equations are also known as **structural equations** [2].

The notation "$pa_i$" refers to the "parents" of $V_i$ and is motivated by the assumption that the model factorizes as a directed graph, here assumed to be a directed acyclic graph (DAG). The model is causal in that, given a distribution $P(U)$ over the background variables $U$, we can derive the distribution of a subset $Z \subseteq V$ following an **intervention** on $V \setminus Z$. An intervention on variable $V_i$ is the substitution of equation $V_i = f_i(pa_i, U_{pa_i})$ with the equation $V_i = v$ for some $v$. This captures the idea of an agent, external to the system, modifying it by forcefully assigning value $v$ to $V_i$, for example as in a randomized experiment.

The specification of $F$ is a strong assumption but allows for the calculation of **counterfactual** quantities. In brief, consider the following counterfactual statement, "the value of $Y$ if $Z$ had taken value $z$", for two observable variables $Z$ and $Y$. By assumption, the state of any observable variable is fully determined by the background variables and structural equations. The counterfactual is modeled as the solution for $Y$ for a given $U = u$ where the equations for $Z$ are replaced with $Z = z$. We denote it by $Y_{Z \leftarrow z}(u)$ [28], and sometimes as $Y_z$ if the context of the notation is clear.

Counterfactual inference, as specified by a causal model $(U, V, F)$ given evidence $W$, is the computation of probabilities $P(Y_{Z \leftarrow z}(U) \mid W = w)$, where $W$, $Z$ and $Y$ are subsets of $V$. Inference proceeds in three steps, as explained in more detail in Chapter 4 of Pearl et al. [29]: 1. **Abduction**: for a given prior on $U$, compute the posterior distribution of $U$ given the evidence $W = w$; 2. **Action**: substitute the equations for $Z$ with the interventional values $z$, resulting in the modified set of equations $F_z$; 3. **Prediction**: compute the implied distribution on the remaining elements of $V$ using $F_z$ and the posterior $P(U \mid W = w)$.

## 3 Counterfactual Fairness

Given a predictive problem with fairness considerations, where $A$, $X$ and $Y$ represent the protected attributes, remaining attributes, and output of interest respectively, let us assume that we are given a causal model $(U, V, F)$, where $V \equiv A \cup X$. We postulate the following criterion for predictors of $Y$.

**Definition 5** (Counterfactual fairness). *Predictor $\hat{Y}$ is **counterfactually fair** if under any context $X = x$ and $A = a$,*

$$P(\hat{Y}_{A \leftarrow a}(U) = y \mid X = x, A = a) = P(\hat{Y}_{A \leftarrow a'}(U) = y \mid X = x, A = a), \qquad (1)$$

*for all $y$ and for any value $a'$ attainable by $A$.*

This notion is closely related to **actual causes** [13], or token causality in the sense that, to be fair, $A$ should not be a cause of $\hat{Y}$ in any individual instance. In other words, changing $A$ while holding things which are not causally dependent on $A$ constant will not change the distribution of $\hat{Y}$. We also emphasize that counterfactual fairness is an individual-level definition. This is substantially different from comparing different individuals that happen to share the same "treatment" $A = a$ and coincide on the values of $X$, as discussed in Section 4.3.1 of [29] and the Supplementary Material. Differences between $X_a$ and $X_{a'}$ must be caused by variations on $A$ only. Notice also that this definition is agnostic with respect to how good a predictor $\hat{Y}$ is, which we discuss in Section 4.

**Relation to individual fairness**. IF is agnostic with respect to its notion of similarity metric, which is both a strength (generality) and a weakness (no unified way of defining similarity). Counterfactuals and similarities are related, as in the classical notion of distances between "worlds" corresponding to different counterfactuals [23]. If $\hat{Y}$ is a deterministic function of $W \subset A \cup X \cup U$, as in several of

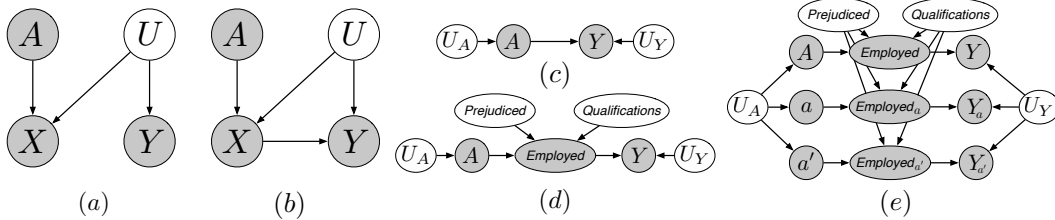

Figure 1: (a), (b) Two causal models for different real-world fair prediction scenarios. See Section 3.1 for discussion. (c) The graph corresponding to a causal model with $A$ being the protected attribute and $Y$ some outcome of interest, with background variables assumed to be independent. (d) Expanding the model to include an intermediate variable indicating whether the individual is employed with two (latent) background variables **Prejudiced** (if the person offering the job is prejudiced) and **Qualifications** (a measure of the individual's qualifications). (e) A twin network representation of this system [28] under two different counterfactual levels for $A$. This is created by copying nodes descending from $A$, which inherit unaffected parents from the factual world.

our examples to follow, then IF can be defined by treating equally two individuals with the same $W$ in a way that is also counterfactually fair.

**Relation to Pearl et al. [29]**. In Example 4.4.4 of [29], the authors condition instead on $X$, $A$, and the observed realization of $\hat{Y}$, and calculate the probability of the counterfactual realization $\hat{Y}_{A \leftarrow a'}$ differing from the factual. This example conflates the predictor $\hat{Y}$ with the outcome $Y$, of which we remain agnostic in our definition but which is used in the construction of $\hat{Y}$ as in Section 4. Our framing makes the connection to machine learning more explicit.

## 3.1 Examples

To provide an intuition for counterfactual fairness, we will consider two real-world fair prediction scenarios: **insurance pricing** and **crime prediction**. Each of these correspond to one of the two causal graphs in Figure 1(a),(b). The Supplementary Material provides a more mathematical discussion of these examples with more detailed insights.

**Scenario 1: The Red Car.** A car insurance company wishes to price insurance for car owners by predicting their accident rate $Y$. They assume there is an unobserved factor corresponding to aggressive driving $U$, that (a) causes drivers to be more likely have an accident, and (b) causes individuals to prefer red cars (the observed variable $X$). Moreover, individuals belonging to a certain race $A$ are more likely to drive red cars. However, these individuals are no more likely to be aggressive or to get in accidents than any one else. We show this in Figure 1(a). Thus, using the red car feature $X$ to predict accident rate $Y$ would seem to be an unfair prediction because it may charge individuals of a certain race more than others, even though no race is more likely to have an accident. Counterfactual fairness agrees with this notion: changing $A$ while holding $U$ fixed will also change $X$ and, consequently, $\hat{Y}$. Interestingly, we can show (Supplementary Material) that in a linear model, regressing $Y$ on $A$ and $X$ is equivalent to regressing on $U$, so off-the-shelf regression here is counterfactually fair. Regressing $Y$ on $X$ alone obeys the FTU criterion but is not counterfactually fair, so *omitting $A$ (FTU) may introduce unfairness into an otherwise fair world*.

**Scenario 2: High Crime Regions.** A city government wants to estimate crime rates by neighborhood to allocate policing resources. Its analyst constructed training data by merging (1) a registry of residents containing their neighborhood $X$ and race $A$, with (2) police records of arrests, giving each resident a binary label with $Y = 1$ indicating a criminal arrest record. Due to historically segregated housing, the location $X$ depends on $A$. Locations $X$ with more police resources have larger numbers of arrests $Y$. And finally, $U$ represents the totality of socioeconomic factors and policing practices that both influence where an individual may live and how likely they are to be arrested and charged. This can all be seen in Figure 1(b).

In this example, higher observed arrest rates in some neighborhoods are due to greater policing there, not because people of different races are any more or less likely to break the law. The label $Y = 0$

does not mean someone has never committed a crime, but rather that they have not been caught. *If individuals in the training data have not already had equal opportunity, algorithms enforcing EO will not remedy such unfairness.* In contrast, a counterfactually fair approach would model differential enforcement rates using $U$ and base predictions on this information rather than on $X$ directly.

In general, we need a multistage procedure in which we first derive latent variables $U$, and then based on them we minimize some loss with respect to $Y$. This is the core of the algorithm discussed next.

### 3.2 Implications

One simple but important implication of the definition of counterfactual fairness is the following:

**Lemma 1.** *Let $\mathcal{G}$ be the causal graph of the given model $(U, V, F)$. Then $\hat{Y}$ will be counterfactually fair if it is a function of the non-descendants of $A$.*

*Proof.* Let $W$ be any non-descendant of $A$ in $\mathcal{G}$. Then $W_{A \leftarrow a}(U)$ and $W_{A \leftarrow a'}(U)$ have the same distribution by the three inferential steps in Section 2.2. Hence, the distribution of any function $\hat{Y}$ of the non-descendants of $A$ is invariant with respect to the counterfactual values of $A$. $\qquad\square$

This does not exclude using a descendant $W$ of $A$ as a possible input to $\hat{Y}$. However, this will only be possible in the case where the overall dependence of $\hat{Y}$ on $A$ disappears, which will not happen in general. Hence, Lemma 1 provides the most straightforward way to achieve counterfactual fairness. In some scenarios, it is desirable to define path-specific variations of counterfactual fairness that allow for the inclusion of some descendants of $A$, as discussed by [21, 27] and the Supplementary Material.

**Ancestral closure of protected attributes.** Suppose that a parent of a member of $A$ is not in $A$. Counterfactual fairness allows for the use of it in the definition of $\hat{Y}$. If this seems counterintuitive, then we argue that the fault should be at the postulated set of protected attributes rather than with the definition of counterfactual fairness, and that typically we should expect set $A$ to be closed under ancestral relationships given by the causal graph. For instance, if *Race* is a protected attribute, and *Mother's race* is a parent of *Race*, then it should also be in $A$.

**Dealing with historical biases and an existing fairness paradox.** The explicit difference between $\hat{Y}$ and $Y$ allows us to tackle historical biases. For instance, let $Y$ be an indicator of whether a client defaults on a loan, while $\hat{Y}$ is the actual decision of giving the loan. Consider the DAG $A \rightarrow Y$, shown in Figure 1(c) with the explicit inclusion of set $U$ of independent background variables. $Y$ is the objectively ideal measure for decision making, the binary indicator of the event that the individual defaults on a loan. If $A$ is postulated to be a protected attribute, then the predictor $\hat{Y} = Y = f_Y(A, U)$ is not counterfactually fair, with the arrow $A \rightarrow Y$ being (for instance) the result of a world that punishes individuals in a way that is out of their control. Figure 1(d) shows a finer-grained model, where the path is mediated by a measure of whether the person is employed, which is itself caused by two background factors: one representing whether the person hiring is prejudiced, and the other the employee's qualifications. In this world, $A$ is a cause of defaulting, even if mediated by other variables[3]. The counterfactual fairness principle however forbids us from using $Y$: using the twin network [4] of Pearl [28], we see in Figure 1(e) that $Y_a$ and $Y_{a'}$ need not be identically distributed given the background variables.

In contrast, any function of variables not descendants of $A$ can be used a basis for fair decision making. This means that any variable $\hat{Y}$ defined by $\hat{Y} = g(U)$ will be counterfactually fair for any function $g(\cdot)$. Hence, given a causal model, the functional defined by the function $g(\cdot)$ minimizing some predictive error for $Y$ will satisfy the criterion, as proposed in Section 4.1. We are essentially learning a projection of $Y$ into the space of fair decisions, removing historical biases as a by-product.

Counterfactual fairness also provides an answer to some problems on the incompatibility of fairness criteria. In particular, consider the following problem raised independently by different authors (e.g.,

[7, 22]), illustrated below for the binary case: ideally, we would like our predictors to obey both Equality of Opportunity and the *predictive parity* criterion defined by satisfying

$$P(Y = 1 \mid \hat{Y} = 1, A = 1) = P(Y = 1 \mid \hat{Y} = 1, A = 0),$$

as well as the corresponding equation for $\hat{Y} = 0$. It has been shown that if $Y$ and $A$ are marginally associated (e.g., recidivism and race are associated) and $Y$ is not a deterministic function of $\hat{Y}$, then the two criteria cannot be reconciled. Counterfactual fairness throws a light in this scenario, suggesting that both EO and predictive parity may be insufficient if $Y$ and $A$ are associated: assuming that $A$ and $Y$ are unconfounded (as expected for demographic attributes), this is the result of $A$ being a cause of $Y$. By counterfactual fairness, we should *not* want to use $Y$ as a basis for our decisions, instead aiming at some function $Y_{\perp A}$ of variables which are not caused by $A$ but are predictive of $Y$. $\hat{Y}$ is defined in such a way that is an estimate of the "closest" $Y_{\perp A}$ to $Y$ according to some preferred risk function. *This makes the incompatibility between EO and predictive parity irrelevant*, as $A$ and $Y_{\perp A}$ will be independent by construction given the model assumptions.

## 4 Implementing Counterfactual Fairness

As discussed in the previous Section, we need to relate $\hat{Y}$ to $Y$ if the predictor is to be useful, and we restrict $\hat{Y}$ to be a (parameterized) function of the non-descendants of $A$ in the causal graph following Lemma 1. We next introduce an algorithm, then discuss assumptions that can be used to express counterfactuals.

### 4.1 Algorithm

Let $\hat{Y} \equiv g_\theta(U, X_{\not\succ A})$ be a predictor parameterized by $\theta$, such as a logistic regression or a neural network, and where $X_{\not\succ A} \subseteq X$ are non-descendants of $A$. Given a loss function $l(\cdot, \cdot)$ such as squared loss or log-likelihood, and training data $\mathcal{D} \equiv \{(A^{(i)}, X^{(i)}, Y^{(i)})\}$ for $i = 1, 2, \ldots, n$, we define $L(\theta) \equiv \sum_{i=1}^{n} \mathbb{E}[l(y^{(i)}, g_\theta(U^{(i)}, x_{\not\succ A}^{(i)})) \mid x^{(i)}, a^{(i)}]/n$ as the empirical loss to be minimized with respect to $\theta$. Each expectation is with respect to random variable $U^{(i)} \sim P_{\mathcal{M}}(U \mid x^{(i)}, a^{(i)})$ where $P_{\mathcal{M}}(U \mid x, a)$ is the conditional distribution of the background variables as given by a causal model $\mathcal{M}$ that is available by assumption. If this expectation cannot be calculated analytically, Markov chain Monte Carlo (MCMC) can be used to approximate it as in the following algorithm.

1:  **procedure** FAIRLEARNING($\mathcal{D}, \mathcal{M}$)                             ▷ Learned parameters $\hat{\theta}$
2:      For each data point $i \in \mathcal{D}$, sample $m$ MCMC samples $U_1^{(i)}, \ldots, U_m^{(i)} \sim P_{\mathcal{M}}(U \mid x^{(i)}, a^{(i)})$.
3:      Let $\mathcal{D}'$ be the augmented dataset where each point $(a^{(i)}, x^{(i)}, y^{(i)})$ in $\mathcal{D}$ is replaced with the corresponding $m$ points $\{(a^{(i)}, x^{(i)}, y^{(i)}, u_j^{(i)})\}$.
4:      $\hat{\theta} \leftarrow \operatorname{argmin}_\theta \sum_{i' \in \mathcal{D}'} l(y^{(i')}, g_\theta(U^{(i')}, x_{\not\succ A}^{(i')}))$.
5:  **end procedure**

At prediction time, we report $\tilde{Y} \equiv \mathbb{E}[\hat{Y}(U^\star, x_{\not\succ A}^\star) \mid x^\star, a^\star]$ for a new data point $(a^\star, x^\star)$.

**Deconvolution perspective.** The algorithm can be understood as a deconvolution approach that, given observables $A \cup X$, extracts its latent sources and pipelines them into a predictive model. We advocate that *counterfactual assumptions must underlie all approaches that claim to extract the sources of variation of the data as "fair" latent components.* As an example, Louizos et al. [24] start from the DAG $A \rightarrow X \leftarrow U$ to extract $P(U \mid X, A)$. As $U$ and $A$ are not independent given $X$ in this representation, a type of penalization is enforced to create a posterior $P_{fair}(U \mid A, X)$ that is close to the model posterior $P(U \mid A, X)$ while satisfying $P_{fair}(U \mid A = a, X) \approx P_{fair}(U \mid A = a', X)$. But *this is neither necessary nor sufficient for counterfactual fairness*. The model for $X$ given $A$ and $U$ must be justified by a causal mechanism, and that being the case, $P(U \mid A, X)$ requires no postprocessing. As a matter of fact, model $\mathcal{M}$ can be learned by penalizing empirical dependence measures between $U$ and $pa_i$ for a given $V_i$ (e.g. Mooij et al. [26]), but this concerns $\mathcal{M}$ and not $\hat{Y}$, and is motivated by explicit assumptions about structural equations, as described next.

## 4.2 Designing the Input Causal Model

Model $\mathcal{M}$ must be provided to algorithm FAIRLEARNING. Although this is well understood, it is worthwhile remembering that causal models always require strong assumptions, even more so when making counterfactual claims [8]. Counterfactuals assumptions such as structural equations are in general unfalsifiable even if interventional data for all variables is available. This is because there are infinitely many structural equations compatible with the same observable distribution [28], be it observational or interventional. Having passed testable implications, the remaining components of a counterfactual model should be understood as conjectures formulated according to the best of our knowledge. Such models should be deemed provisional and prone to modifications if, for example, new data containing measurement of variables previously hidden contradict the current model.

We point out that we do not need to specify a fully deterministic model, and structural equations can be relaxed as conditional distributions. In particular, the concept of counterfactual fairness holds under three levels of assumptions of increasing strength:

**Level 1.** Build $\hat{Y}$ using only the observable non-descendants of $A$. This only requires partial causal ordering and no further causal assumptions, but in many problems there will be few, if any, observables which are not descendants of protected demographic factors.

**Level 2.** Postulate background latent variables that act as non-deterministic causes of observable variables, based on explicit domain knowledge and learning algorithms[5]. Information about $X$ is passed to $\hat{Y}$ via $P(U \mid x, a)$.

**Level 3.** Postulate a fully deterministic model with latent variables. For instance, the distribution $P(V_i \mid pa_i)$ can be treated as an additive error model, $V_i = f_i(pa_i) + e_i$ [31]. The error term $e_i$ then becomes an input to $\hat{Y}$ as calculated from the observed variables. This maximizes the information extracted by the fair predictor $\hat{Y}$.

## 4.3 Further Considerations on Designing the Input Causal Model

One might ask what we can lose by defining causal fairness measures involving only non-counterfactual causal quantities, such as enforcing $P(\hat{Y} = 1 \mid do(A = a)) = P(\hat{Y} = 1 \mid do(A = a'))$ instead of our counterfactual criterion. The reason is that the above equation is only a constraint on an average effect. Obeying this criterion provides no guarantees against, for example, having half of the individuals being strongly "negatively" discriminated and half of the individuals strongly "positively" discriminated. We advocate that, for fairness, society should not be satisfied in pursuing only counterfactually-free guarantees. While one may be willing to claim posthoc that the equation above masks no balancing effect so that individuals receive approximately the same distribution of outcomes, *that itself is just a counterfactual claim in disguise.* Our approach is to make counterfactual assumptions explicit. When unfairness is judged to follow only some "pathways" in the causal graph (in a sense that can be made formal, see [21, 27]), nonparametric assumptions about the independence of counterfactuals may suffice, as discussed by [27]. In general, nonparametric assumptions may not provide identifiable adjustments even in this case, as also discussed in our Supplementary Material. If competing models with different untestable assumptions are available, there are ways of simultaneously enforcing a notion of approximate counterfactual fairness in all of them, as introduced by us in [32]. Other alternatives include exploiting bounds on the contribution of hidden variables [29, 33].

Another issue is the interpretation of causal claims involving demographic variables such as race and sex. Our view is that such constructs are the result of translating complex events into random variables and, despite some controversy, we consider counterproductive to claim that e.g. race and sex cannot be causes. An idealized intervention on some $A$ at a particular time can be seen as a notational shortcut to express a conjunction of more specific interventions, which may be individually doable but jointly impossible in practice. It is the plausibility of complex, even if impossible to practically manipulate, causal chains from $A$ to $Y$ that allows us to claim that unfairness is real [11]. Experiments for constructs exist, such as randomizing names in job applications to make them race-blind. They do not contradict the notion of race as a cause, and can be interpreted as an intervention on a particular aspect of the construct "race," such as "race perception" (e.g. Section 4.4.4 of [29]).

# 5  Illustration: Law School Success

We illustrate our approach on a practical problem that requires fairness, the *prediction of success in law school*. A second problem, *understanding the contribution of race to police stops*, is described in the Supplementary Material. Following closely the usual framework for assessing causal models in the machine learning literature, the goal of this experiment is to quantify how our algorithm behaves with finite sample sizes while assuming ground truth compatible with a synthetic model.

**Problem definition: Law school success**

The Law School Admission Council conducted a survey across 163 law schools in the United States [35]. It contains information on 21,790 law students such as their entrance exam scores (LSAT), their grade-point average (GPA) collected prior to law school, and their first year average grade (FYA).

Given this data, a school may wish to predict if an applicant will have a high FYA. The school would also like to make sure these predictions are not biased by an individual's race and sex. However, the LSAT, GPA, and FYA scores, may be biased due to social factors. We compare our framework with two unfair baselines: 1. **Full**: the standard technique of using all features, including sensitive features such as race and sex to make predictions; 2. **Unaware**: fairness through unawareness, where we do not use race and sex as features. For comparison, we generate predictors $\hat{Y}$ for all models using logistic regression.

**Fair prediction.**  As described in Section 4.2, there are three ways in which we can model a counterfactually fair predictor of FYA. Level 1 uses any features which are not descendants of race and sex for prediction. Level 2 models latent 'fair' variables which are parents of observed variables. These variables are independent of both race and sex. Level 3 models the data using an additive error model, and uses the independent error terms to make predictions. These models make increasingly strong assumptions corresponding to increased predictive power. We split the dataset 80/20 into a train/test set, preserving label balance, to evaluate the models.

As we believe LSAT, GPA, and FYA are all biased by race and sex, we cannot use any observed features to construct a counterfactually fair predictor as described in Level 1.

In Level 2, we postulate that a latent variable: a student's **knowledge** (K), affects GPA, LSAT, and FYA scores. The causal graph corresponding to this model is shown in Figure 2, (**Level 2**). This is a short-hand for the distributions:

$$\text{GPA} \sim \mathcal{N}(b_G + w_G^K K + w_G^R R + w_G^S S, \sigma_G), \qquad \text{FYA} \sim \mathcal{N}(w_F^K K + w_F^R R + w_F^S S, 1),$$
$$\text{LSAT} \sim \text{Poisson}(\exp(b_L + w_L^K K + w_L^R R + w_L^S S)), \qquad \text{K} \sim \mathcal{N}(0,1)$$

We perform inference on this model using an observed training set to estimate the posterior distribution of $K$. We use the probabilistic programming language Stan [34] to learn $K$. We call the predictor constructed using $K$, **Fair** $K$.

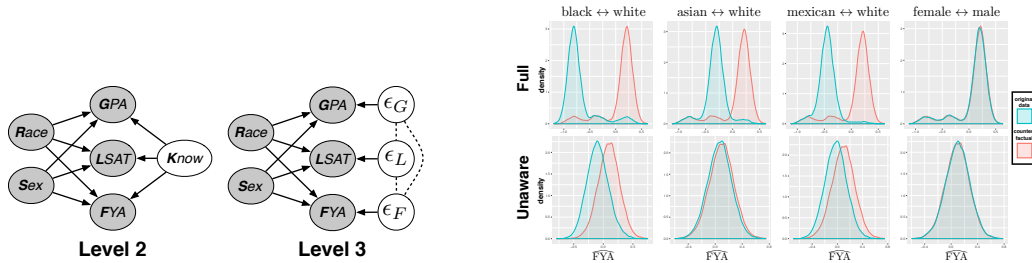

Figure 2: **Left:** A causal model for the problem of predicting law school success fairly. **Right:** Density plots of predicted $\text{FYA}_a$ and $\text{FYA}_{a'}$.

In Level 3, we model GPA, LSAT, and FYA as continuous variables with additive error terms independent of race and sex (that may in turn be correlated with one-another). This model is shown

Table 1: Prediction results using logistic regression. Note that we must sacrifice a small amount of accuracy to ensuring counterfactually fair prediction (Fair $K$, Fair Add), versus the models that use unfair features: GPA, LSAT, race, sex (Full, Unaware).

|  | Full | Unaware | Fair $K$ | Fair Add |
|---|---|---|---|---|
| RMSE | 0.873 | 0.894 | 0.929 | 0.918 |

in Figure 2, (**Level 3**), and is expressed by:

$$\text{GPA} = b_G + w_G^R R + w_G^S S + \epsilon_G, \quad \epsilon_G \sim p(\epsilon_G)$$
$$\text{LSAT} = b_L + w_L^R R + w_L^S S + \epsilon_L, \quad \epsilon_L \sim p(\epsilon_L)$$
$$\text{FYA} = b_F + w_F^R R + w_F^S S + \epsilon_F, \quad \epsilon_F \sim p(\epsilon_F)$$

We estimate the error terms $\epsilon_G, \epsilon_L$ by first fitting two models that each use race and sex to individually predict GPA and LSAT. We then compute the residuals of each model (e.g., $\epsilon_G = \text{GPA} - \hat{Y}_{\text{GPA}}(R, S)$). We use these residual estimates of $\epsilon_G, \epsilon_L$ to predict FYA. We call this *Fair Add*.

**Accuracy.** We compare the RMSE achieved by logistic regression for each of the models on the test set in Table 1. The **Full** model achieves the lowest RMSE as it uses race and sex to more accurately reconstruct FYA. Note that in this case, this model is not fair even if the data was generated by one of the models shown in Figure 2 as it corresponds to Scenario 3. The (also unfair) **Unaware** model still uses the unfair variables GPA and LSAT, but because it does not use race and sex it cannot match the RMSE of the **Full** model. As our models satisfy counterfactual fairness, they trade off some accuracy. Our first model **Fair $K$** uses weaker assumptions and thus the RMSE is highest. Using the Level 3 assumptions, as in **Fair Add** we produce a counterfactually fair model that trades slightly stronger assumptions for lower RMSE.

**Counterfactual fairness.** We would like to empirically test whether the baseline methods are counterfactually fair. To do so we will assume the true model of the world is given by Figure 2, (**Level 2**). We can fit the parameters of this model using the observed data and evaluate counterfactual fairness by sampling from it. Specifically, we will generate samples from the model given either the observed race and sex, or *counterfactual* race and sex variables. We will fit models to both the original and counterfactual sampled data and plot how the distribution of predicted FYA changes for both baseline models. Figure 2 shows this, where each row corresponds to a baseline predictor and each column corresponds to the counterfactual change. In each plot, the blue distribution is density of predicted FYA for the original data and the red distribution is this density for the counterfactual data. If a model is counterfactually fair we would expect these distributions to lie exactly on top of each other. Instead, we note that the **Full** model exhibits counterfactual unfairness for all counterfactuals except sex. We see a similar trend for the **Unaware** model, although it is closer to being counterfactually fair. To see why these models seem to be fair w.r.t. to sex we can look at weights of the DAG which generates the counterfactual data. Specifically the DAG weights from (male,female) to GPA are (0.93,1.06) and from (male,female) to LSAT are (1.1,1.1). Thus, these models are fair w.r.t. to sex simply because of a very weak causal link between sex and GPA/LSAT.

## 6 Conclusion

We have presented a new model of fairness we refer to as *counterfactual fairness*. It allows us to propose algorithms that, rather than simply ignoring protected attributes, are able to take into account the different social biases that may arise towards individuals based on ethically sensitive attributes and compensate for these biases effectively. We experimentally contrasted our approach with previous fairness approaches and show that our explicit causal models capture these social biases and make clear the implicit trade-off between prediction accuracy and fairness in an unfair world. We propose that fairness should be regulated by explicitly modeling the causal structure of the world. Criteria based purely on probabilistic independence cannot satisfy this and are unable to address *how* unfairness is occurring in the task at hand. By providing such causal tools for addressing fairness questions we hope we can provide practitioners with customized techniques for solving a wide array of fairness modeling problems.

**Acknowledgments**

This work was supported by the Alan Turing Institute under the EPSRC grant EP/N510129/1. CR acknowledges additional support under the EPSRC Platform Grant EP/P022529/1. We thank Adrian Weller for insightful feedback, and the anonymous reviewers for helpful comments.

## Footnotes

*Equal contribution. This work was done while JL was a Research Fellow at the Alan Turing Institute.

[2]https://obamawhitehouse.archives.gov/blog/2016/05/04/big-risks-big-opportunities-intersection-big-data-and-civil-rights

[3]For example, if the function determining employment $f_E(A, P, Q) \equiv I_{(Q>0, P=0 \text{ or } A \neq a)}$ then an individual with sufficient qualifications and prejudiced potential employer may have a different counterfactual employment value for $A = a$ compared to $A = a'$, and a different chance of default.

[4]In a nutshell, this is a graph that simultaneously depicts "multiple worlds" parallel to the factual realizations. In this graph, all multiple worlds share the same background variables, but with different consequences in the remaining variables depending on which counterfactual assignments are provided.

[5]In some domains, it is actually common to build a model entirely around latent constructs with few or no observable parents nor connections among observed variables [2].

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
