[Supplementary Material]

## S1 Population Level vs Individual Level Causal Effects

As discussed in Section 3, counterfactual fairness is an individual-level definition. This is fundamentally different from comparing different units that happen to share the same "treatment" $A = a$ and coincide on the values of $X$. To see in detail what this means, consider the following thought experiment.

Let us assess the causal effect of $A$ on $\hat{Y}$ by controlling $A$ at two levels, $a$ and $a'$. In Pearl's notation, where "$do(A = a)$" expresses an intervention on $A$ at level $a$, we have that

$$\mathbb{E}[\hat{Y} \mid do(A = a), X = x] - \mathbb{E}[\hat{Y} \mid do(A = a'), X = x], \tag{2}$$

is a measure of causal effect, sometimes called the average causal effect (ACE). It expresses the change that is expected when we intervene on $A$ while observing the attribute set $X = x$, under two levels of treatment. If this effect is non-zero, $A$ is considered to be a cause of $\hat{Y}$.

This raises a subtlety that needs to be addressed: in general, this effect will be non-zero *even if $\hat{Y}$ is counterfactually fair*. This may sound counter-intuitive: protected attributes such as race and gender are causes of our counterfactually fair decisions.

In fact, this is not a contradiction, as the ACE in Equation (2) is different from counterfactual effects. The ACE contrasts two independent exchangeable units of the population, and it is a perfectly valid way of performing decision analysis. However, the value of $X = x$ is affected by different background variables corresponding to different individuals. That is, the causal effect (2) contrasts two units that receive different treatments but which happen to coincide on $X = x$. To give a synthetic example, imagine the simple structural equation

$$X = A + U.$$

The ACE quantifies what happens among people with $U = x - a$ against people with $U' = x - a'$. If, for instance, $\hat{Y} = \lambda U$ for $\lambda \neq 0$, then the effect (2) is $\lambda(a - a') \neq 0$.

Contrary to that, the counterfactual difference is zero. That is,

$$\mathbb{E}[\hat{Y}_{A \leftarrow a}(U) \mid A = a, X = x] - \mathbb{E}[\hat{Y}_{A \leftarrow a'}(U) \mid A = a, X = x] = \lambda U - \lambda U = 0.$$

In another perspective, we can interpret the above just as if we had *measured $U$* from the beginning rather than performing abduction. We then generate $\hat{Y}$ from some $g(U)$, so $U$ is the within-unit cause of $\hat{Y}$ and not $A$.

If $U$ cannot be deterministically derived from $\{A = a, X = x\}$, the reasoning is similar. By abduction, the distribution of $U$ will typically depend on $A$, and hence so will $\hat{Y}$ when marginalizing over $U$. Again, this seems to disagree with the intuition that our predictor should be not be caused by $A$. However, this once again is a comparison *across individuals*, not within an individual.

It is this balance among $(A, X, U)$ that explains, in the examples of Section 3.1, why some predictors are counterfactually fair even though they are functions of the same variables $\{A, X\}$ used by unfair predictors: such functions must correspond to particular ways of balancing the observables that, by way of the causal assumptions, cancel out the effect of $A$.

**More on conditioning and alternative definitions.** As discussed in Example 4.4.4 of Pearl et al. [29], a different proposal for assessing fairness can be defined via the following concept:

**Definition 6** (Probability of sufficiency). *We define the probability of event $\{A = a\}$ being a sufficient cause for our decision $\hat{Y}$, contrasted against $\{A = a'\}$, as*

$$P(\hat{Y}_{A \leftarrow a'}(U) \neq y \mid X = x, A = a, \hat{Y} = y). \tag{3}$$

We can then, for instance, claim that $\hat{Y}$ is a fair predictor if this probability is below some pre-specified bound for all $(x, a, a')$. The shortcomings of this definition come from its original motivation: to *explain* the behavior of an *existing* decision protocol, where $\hat{Y}$ is the current practice and which in a unclear way is conflated with $Y$. The implication is that if $\hat{Y}$ is to be designed instead of being a natural measure of existing behaviour, then we are using $\hat{Y}$ itself as evidence for the background

variables $U$. This does not make sense if $\hat{Y}$ is yet to be designed by us. If $\hat{Y}$ is to be interpreted as $Y$, then this does not provide a clear recipe on how to build $\hat{Y}$: while we can use $Y$ to learn a causal model, we cannot use it to collect training data evidence for $U$ *as the outcome $Y$ will not be available to us at prediction time*. For this reason, we claim that while probability of sufficiency is useful as a way of assessing an existing decision making process, it is not as natural as counterfactual fairness in the context of machine learning.

**Approximate fairness and model validation.** The notion of probability of sufficiency raises the question on how to define approximate, or high probability, counterfactual fairness. This is an important question that we address in [32]. Before defining an approximation, it is important to first expose in detail what the exact definition is, which is the goal of this paper.

We also do not address the validation of the causal assumptions used by the input causal model of the FAIRLEARNING algorithm in Section 4.1. The reason is straightforward: this validation is an entirely self-contained step of the implementation of counterfactual fairness. An extensive literature already exists in this topic which the practitioner can refer to (a classic account for instance is [3]), and which can be used as-is in our context.

The experiments performed in Section 5 can be criticized by the fact that they rely on a model that obeys our assumptions, and "obviously" our approach should work better than alternatives. This criticism is not warranted: in machine learning, causal inference is typically assessed through simulations which assume that the true model lies in the family covered by the algorithm. Algorithms, including FAIRLEARNING, are justified in the population sense. How different competitors behave with finite sample sizes is the primary question to be studied in an empirical study of a new concept, where we control for the correctness of the assumptions. Although sensitivity analysis is important, there are many degrees of freedom on how this can be done. Robustness issues are better addressed by extensions focusing on approximate versions of counterfactual fairness. This will be covered in later work.

**Stricter version.** For completeness of exposition, notice that the definition of counterfactual fairness could be strengthened to

$$P(\hat{Y}_{A \leftarrow a}(U) = \hat{Y}_{A \leftarrow a'}(U) \mid X = x, A = a) = 1. \tag{4}$$

This is different from the original definition in the case where $\hat{Y}(U)$ is a random variable with a different source of randomness for different counterfactuals (for instance, if $\hat{Y}$ is given by some black-box function of $U$ with added noise that is independent across each countefactual value of $A$). In such a situation, the event $\{\hat{Y}_{A \leftarrow a}(U) = \hat{Y}_{A \leftarrow a'}(U)\}$ will itself have probability zero even if $P(\hat{Y}_{A \leftarrow a}(U) = y \mid X = x, A = a) = P(\hat{Y}_{A \leftarrow a'}(U) = y \mid X = x, A = a)$ for all $y$. We do not consider version (4) as in our view it does not feel as elegant as the original, and it is also unclear whether adding an independent source of randomness fed to $\hat{Y}$ would itself be considered unfair. Moreover, if $\hat{Y}(U)$ is assumed to be a deterministic function of $U$ and $X$, as in FAIRLEARNING, then the two definitions are the same[6]. Informally, this stricter definition corresponds to a notion of "almost surely equality" as opposed to "equality in distribution." Without assuming that $\hat{Y}$ is a deterministic function of $U$ and $X$, even the stricter version does not protect us against measure zero events where the counterfactuals are different. The definition of counterfactual fairness concisely emphasizes that $U$ can be a random variable, and clarifies which conditional distribution it follows. Hence, it is our preferred way of introducing the concept even though it does not explicit suggests whether $\hat{Y}(U)$ has random inputs besides $U$.

## S2 Relation to Demographic Parity

Consider the graph $A \rightarrow X \rightarrow Y$. In general, if $\hat{Y}$ is a function of $X$ only, then $\hat{Y}$ need not obey demographic parity, i.e.

$$P(\hat{Y} \mid A = a) \neq P(\hat{Y} \mid A = a'),$$

where, since $\hat{Y}$ is a function of $X$, the probabilities are obtained by marginalizing over $P(X \mid A = a)$ and $P(X \mid A = a')$, respectively.

If we postulate a structural equation $X = \alpha A + e_X$, then given $A$ and $X$ we can deduce $e_X$. If $\hat{Y}$ is a function of $e_X$ only and, by assumption, $e_X$ is marginally independent of $A$, then $\hat{Y}$ is marginally independent of $A$: this follows the interpretation given in the previous section, where we interpret $e_X$ as "known" despite being mathematically deduced from the observation $(A = a, X = x)$. Therefore, the assumptions imply that $\hat{Y}$ will satisfy demographic parity, and that can be falsified. By way of contrast, if $e_X$ is not uniquely identifiable from the structural equation and $(A, X)$, then the distribution of $\hat{Y}$ depends on the value of $A$ as we marginalize $e_X$, and demographic parity will not follow. This leads to the following:

**Lemma 2.** *If all background variables $U' \subseteq U$ in the definition of $\hat{Y}$ are determined from $A$ and $X$, and all observable variables in the definition of $\hat{Y}$ are independent of $A$ given $U'$, then $\hat{Y}$ satisfies demographic parity.*

Thus, counterfactual fairness can be thought of as a counterfactual analog of demographic parity, as present in the Red Car example further discussed in the next section.

## S3 Examples Revisited

In Section 3.1, we discussed two examples. We reintroduce them here briefly, add a third example, and explain some consequences of their causal structure to the design of counterfactually fair predictors.

**Scenario 1: The Red Car Revisited.**   In that scenario, the structure $A \to X \leftarrow U \to Y$ implies that $\hat{Y}$ should not use either $X$ or $A$. On the other hand, it is acceptable to use $U$. It is interesting to realize, however, that since $U$ is related to $A$ and $X$, there will be some association between $Y$ and $\{A, X\}$ as discussed in Section S1. In particular, if the structural equation for $X$ is linear, then $U$ is a linear function of $A$ and $X$, and as such $\hat{Y}$ will also be a function of both $A$ and $X$. This is not a problem, as it is still the case that the model implies that this is merely a functional dependence that disappears by conditioning on a postulated latent attribute $U$. Surprisingly, we must make $\hat{Y}$ a indirect function of $A$ if we want a counterfactually fair predictor, as shown in the following Lemma.

**Lemma 3.** *Consider a linear model with the structure in Figure 1(a). Fitting a linear predictor to $X$ only is not counterfactually fair, while the same algorithm will produce a fair predictor using both $A$ and $X$.*

*Proof.* As in the definition, we will consider the population case, where the joint distribution is known. Consider the case where the equations described by the model in Figure 1(a) are deterministic and linear:

$$X = \alpha A + \beta U, \quad Y = \gamma U.$$

Denote the variance of $U$ as $v_U$, the variance of $A$ as $v_A$, and assume all coefficients are non-zero. The predictor $\hat{Y}(X)$ defined by least-squares regression of $Y$ on *only* $X$ is given by $\hat{Y}(X) \equiv \lambda X$, where $\lambda = Cov(X, Y)/Var(X) = \beta \gamma v_U / (\alpha^2 v_A + \beta^2 v_U) \neq 0$. This predictor follows the concept of fairness through unawareness.

We can test whether a predictor $\hat{Y}$ is counterfactually fair by using the procedure described in Section 2.2:

*(i)* Compute $U$ given observations of $X, Y, A$; *(ii)* Substitute the equations involving $A$ with an interventional value $a'$; *(iii)* Compute the variables $X, Y$ with the interventional value $a'$. It is clear here that $\hat{Y}_a(U) = \lambda(\alpha a + \beta U) \neq \hat{Y}_{a'}(U)$. This predictor is not counterfactually fair. Thus, in this case fairness through unawareness actually perpetuates unfairness.

Consider instead doing least-squares regression of $Y$ on $X$ *and* $A$. Note that $\hat{Y}(X, A) \equiv \lambda_X X + \lambda_A A$ where $\lambda_X, \lambda_A$ can be derived as follows:

$$\begin{pmatrix} \lambda_X \\ \lambda_A \end{pmatrix} = \begin{pmatrix} Var(X) & Cov(A,X) \\ Cov(X,A) & Var(A) \end{pmatrix}^{-1} \begin{pmatrix} Cov(X,Y) \\ Cov(A,Y) \end{pmatrix}$$

$$= \frac{1}{\beta^2 v_U v_A} \begin{pmatrix} v_A & -\alpha v_A \\ -\alpha v_A & \alpha^2 v_A + \beta^2 v_U \end{pmatrix} \begin{pmatrix} \beta \gamma v_U \\ 0 \end{pmatrix}$$

$$= \begin{pmatrix} \frac{\gamma}{\beta} \\ \frac{-\alpha \gamma}{\beta} \end{pmatrix} \tag{5}$$

Now imagine we have observed $A = a$. This implies that $X = \alpha a + \beta U$ and our predictor is $\hat{Y}(X,a) = \frac{\gamma}{\beta}(\alpha a + \beta U) + \frac{-\alpha\gamma}{\beta}a = \gamma U$. Thus, if we substitute $a$ with a counterfactual $a'$ (the action step described in Section 2.2) the predictor $\hat{Y}(X,A)$ is unchanged. This is because our predictor is constructed in such a way that any change in $X$ caused by a change in $A$ is cancelled out by the $\lambda_A$. Thus this predictor is counterfactually fair. $\square$

Note that if Figure 1(a) is the true model for the real world then $\hat{Y}(X,A)$ will also satisfy demographic parity and equality of opportunity as $\hat{Y}$ will be unaffected by $A$.

The above lemma holds in a more general case for the structure given in Figure 1(a): any non-constant estimator that depends only on $X$ is not counterfactually fair as changing $A$ always alters $X$.

**Scenario 2: High Crime Regions Revisited.** The causal structure differs from the previous example by the extra edge $X \to Y$. For illustration purposes, assume again that the model is linear. Unlike the previous case, a predictor $\hat{Y}$ trained using $X$ and $A$ is not counterfactually fair. The only change from Scenario 1 is that now $Y$ depends on $X$ as follows: $Y = \gamma U + \theta X$. Now if we solve for $\lambda_X, \lambda_A$ it can be shown that $\hat{Y}(X,a) = (\gamma - \frac{\alpha^2 \theta v_A}{\beta v_U})U + \alpha\theta a$. As this predictor depends on the values of $A$ that are not explained by $U$, then $\hat{Y}(X,a) \neq \hat{Y}(X,a')$ and thus $\hat{Y}(X,A)$ is not counterfactually fair.

The following extra example complements the previous two examples.

**Scenario 3: University Success.** A university wants to know if students will be successful post-graduation $Y$. They have information such as: grade point average (GPA), advanced placement (AP) exams results, and other academic features $X$. The university believes however, that an individual's gender $A$ may influence these features and their post-graduation success $Y$ due to social discrimination. They also believe that independently, an individual's latent talent $U$ causes $X$ and $Y$. The structure is similar to Figure 1(a), with the extra edge $A \to Y$. We can again ask, is the predictor $\hat{Y}(X,A)$ counterfactually fair? In this case, the different between this and Scenario 1 is that $Y$ is a function of $U$ and $A$ as follows: $Y = \gamma U + \eta A$. We can again solve for $\lambda_X, \lambda_A$ and show that $\hat{Y}(X,a) = (\gamma - \frac{\alpha\eta v_A}{\beta v_U})U + \eta a$. Again $\hat{Y}(X,A)$ is a function of $A$ not explained by $U$, so it cannot be counterfactually fair.

## S4 Analysis of Individual Pathways

By way of an example, consider the following adaptation of the scenario concerning claims of gender bias in UC Berkeley's admission process in the 1970s, commonly used a textbook example of Simpson's Paradox. For each candidate student's application, we have $A$ as a binary indicator of whether the applicant is female, $X$ as the choice of course to apply for, and $Y$ a binary indicator of whether the application was successful or not. Let us postulate the causal graph that includes the edges $A \to X$ and $X \to Y$ only. We observe that $A$ and $Y$ are negatively associated, which in first instance might suggest discrimination, as gender is commonly accepted here as a protected attribute for college admission. However, in the postulated model it turns out that $A$ and $Y$ are causally independent given $X$. More specifically, women tend to choose more competitive courses (those with higher rejection rate) than men when applying. Our judgment is that the higher rejection among female than male applicants is acceptable, if the mechanism $A \to X$ is interpreted as a choice which is under the control of the applicant. That is, free-will overrides whatever possible cultural background conditions that led to this discrepancy. In the framework of counterfactual fairness, we

could claim that $A$ is not a protected attribute to begin with once we understand how the world works, and that including $A$ in the predictor of success is irrelevant anyway once we include $X$ in the classifier.

However, consider the situation where there is an edge $A \rightarrow Y$, interpreted purely as the effect of discrimination after causally controlling for $X$. While it is now reasonable to postulate $A$ to be a protected attribute, we can still judge that $X$ is not an unfair outcome: there is no need to "deconvolve" $A$ out of $X$ to obtain an estimate of the other causes $U_X$ in the $A \rightarrow X$ mechanism. This suggests a simple modification of the definition of counterfactual fairness. First, given the causal graph $\mathcal{G}$ assumed to encode the causal relationships in our system, define $\mathcal{P}_{\mathcal{G}_A}$ as the set of all directed paths from $A$ to $Y$ in $\mathcal{G}$ which are postulated to correspond to all unfair chains of events where $A$ causes $Y$. Let $X_{\mathcal{P}_{\mathcal{G}_A}^c} \subseteq X$ be the subset of covariates not present in any path in $\mathcal{P}_{\mathcal{G}_A}$. Also, for any vector $x$, let $x_s$ represent the corresponding subvector indexed by $S$. The corresponding uppercase version $X_S$ is used for random vectors.

**Definition 7** ((Path-dependent) counterfactual fairness). *Predictor $\hat{Y}$ is* **(path-dependent) counterfactually fair** *with respect to path set $\mathcal{P}_{\mathcal{G}_A}$ if under any context $X = x$ and $A = a$,*

$$P(\hat{Y}_{A \leftarrow a, X_{\mathcal{P}_{\mathcal{G}_A}^c} \leftarrow x_{\mathcal{P}_{\mathcal{G}_A}^c}}(U) = y \mid X = x, A = a) =$$

$$P(\hat{Y}_{A \leftarrow a', X_{\mathcal{P}_{\mathcal{G}_A}^c} \leftarrow x_{\mathcal{P}_{\mathcal{G}_A}^c}}(U) = y \mid X = x, A = a), \quad (6)$$

*for all $y$ and for any value $a'$ attainable by $A$.*

This notion is related to *controlled direct effects* [29], where we intervene on some paths from $A$ to $Y$, but not others. Paths in $\mathcal{P}_{\mathcal{G}_A}$ are considered here to be the "direct" paths, and we condition on $X$ and $A$ similarly to the definition of probability of sufficiency (3). This definition is the same as the original counterfactual fairness definition for the case where $\mathcal{P}_{\mathcal{G}_A}^c = \emptyset$. Its interpretation is analogous to the original, indicating that for any $X_0 \in X_{\mathcal{P}_{\mathcal{G}_A}^c}$ we are allowed to propagate information from the factual assigment $A = a$, along with what we learned about the background causes $U_{X_0}$, in order to reconstruct $X_0$. The contribution of $A$ is considered acceptable in this case and does not need to be "deconvolved." The implication is that any member of $X_{\mathcal{P}_{\mathcal{G}_A}^c}$ can be included in the definition of $\hat{Y}$. In the example of college applications, we are allowed to use the choice of course $X$ even though $A$ is a confounder for $X$ and $Y$. We are still not allowed to use $A$ directly, bypassing the background variables.

As discussed by [27], there are some counterfactual manipulations usable in a causal definition of fairness that can be performed by exploiting only independence constraints among the counterfactuals: that is, without requiring the explicit description of structural equations or other models for latent variables. A contrast between the two approaches is left for future work, although we stress that they are in some sense complementary: we are motivated mostly by problems such as the one in Figure 1(d), where many of the mediators themselves are considered to be unfairly affected by the protected attribute, and independence constraints among counterfactuals alone are less likely to be useful in identifying constraints for the fitting of a fair predictor.

## S5 The Multifaceted Dynamics of Fairness

One particularly interesting question was raised by one of the reviewers: what is the effect of continuing discrimination after fair decisions are made? For instance, consider the case where banks enforce a fair allocation of loans for business owners regardless of, say, gender. This does not mean such businesses will thrive at a balanced rate if customers continue to avoid female owned business at a disproportionate rate for unfair reasons. Is there anything useful that can be said about this issue from a causal perspective?

The work here proposed regards only what we can influence by changing how machine learning-aided decision making takes place at specific problems. It cannot change directly how society as a whole carry on with their biases. Ironically, it may sound unfair to banks to enforce the allocation of resources to businesses at a rate that does not correspond to the probability of their respective success, even if the owners of the corresponding businesses are not to be blamed by that. One way of conciliating the different perspectives is by modeling how a fair allocation of loans, even if it does not come without a cost, can nevertheless increase the proportion of successful female businesses

Figure 3: A causal model for the stop and frisk dataset.

compared to the current baseline. This change can by itself have an indirect effect on the culture and behavior of a society, leading to diminishing continuing discrimination by a feedback mechanism, as in affirmative action. We believe that in the long run isolated acts of fairness are beneficial even if we do not have direct control on all sources of unfairness in any specific problem. Causal modeling can help on creating arguments about the long run impact of individual contributions as e.g. a type of macroeconomic assessment. There are many challenges, and we should not pretend that precise answers can be obtained, but in theory we should aim at educated quantitative assessments validating how a systemic improvement in society can emerge from localized ways of addressing fairness.

## S6 Case Study: NYC Stop-and-Frisk Data

Since 2002, the New York Police Department (NYPD) has recorded information about every time a police officer has stopped someone. The officer records information such as if the person was searched or frisked, if a weapon was found, their appearance, whether an arrest was made or a summons issued, if force was used, etc. We consider the data collected on males stopped during 2014 which constitutes 38,609 records. We limit our analysis to looking at just males stopped as this accounts for more than 90% of the data. We fit a model which postulates that police interactions is caused by race and a single latent factor labeled *Criminality* that is meant to index other aspects of the individual that have been used by the police and which are independent of race. We do not claim that this model has a solid theoretical basis, we use it below as an illustration on how to carry on an analysis of counterfactually fair decisions. We also describe a spatial analysis of the estimated latent factors.

**Model.** We model this stop-and-frisk data using the graph in Figure 4. Specifically, we posit main causes for the observations: *Arrest* (if an individual was arrested), *Force* (some sort of force was used during the stop), *Frisked*, and *Searched*. The first cause of these observations is some measure of an individual's latent *Criminality*, which we do not observe. We believe that *Criminality* also directly affects *Weapon* (an individual was found to be carrying a weapon). For all of the features previously mentioned we believe there is an additional cause, an individual's *Race* which we do observe. This factor is introduced as we believe that these observations may be biased based on an officer's perception of whether an individual is likely a criminal or not, affected by an individual's *Race*. Thus note that, in this model, *Criminality* is counterfactually fair for the prediction of any characteristic of the individual for problems where *Race* is a protected attribute.

**Visualization on a map of New York City.** Each of the stops can be mapped to longitude and latitude points for where the stop occurred[7]. This allows us to visualize the distribution of two distinct populations: the stops of White and Black Hispanic individuals, shown in Figure 5. We note that there are more White individuals stopped (4492) than Black Hispanic individuals (2414). However, if we look at the arrest distribution (visualized geographically in the second plot) the rate of arrest for White individuals is lower (12.1%) than for Black Hispanic individuals (19.8%, the highest rate for any race in the dataset). Given our model we can ask: "If every individual had been White,

Figure 4: How race affects arrest. The above maps show how altering one's race affects whether or not they will be arrested, according to the model. The left-most plot shows the distribution of White and Black Hispanic populations in the stop-and-frisk dataset. The second plot shows the true arrests for all of the stops. Given our model we can compute whether or not every individual in the dataset would be arrest *had they been white*. We show this counterfactual in the third plot. Similarly, we can compute this counterfactual if everyone had been Black Hispanic, as shown in the fourth plot.

would they have been arrested?". The answer to this is in the third plot. We see that the overall number of arrests decreases (from 5659 to 3722). What if every individual had been Black Hispanic? The fourth plot shows an increase in the number of arrests had individuals been Black Hispanic, according to the model (from 5659 to 6439). The yellow and purple circles show two regions where the difference in counterfactual arrest rates is particularly striking. Thus, the model indicates that, even when everything else in the model is held constant, race has a differential affect on arrest rate under the (strong) assumptions of the model.

## Footnotes

[6]Notice that $\hat{Y}(U)$ is itself a random variable if $U$ is, but the source of randomness, $U$, is the same across all counterfactuals.

[7]https://github.com/stablemarkets/StopAndFrisk