[Reviews · NeurIPS 2017]

Reviewer 1



This paper presents an interesting and valuable contribution to the small but growing literature on fairness in machine learning. Specifically, it provides at least three contributions: (1) a definition of counter factual fairness; (2) an algorithm for learning a model under counter factual fairness; and (3) experiments with that algorithm. The value and convincingness of each of these contributions declines steadily. The value of the contributions of the current paper is sufficient for acceptance, though significant improvements could be made in the clarity of exposition of the algorithm and the extent of the experimentation with the algorithm. Section 4.2 outlines why is is likely that very strong assumptions will need to be made to effectively estimate a model of Y under counterfactual fairness. The assumptions (and the implied analysis techniques) suggest conclusions that will not be particularly robust to violations of those assumptions. This implies a need for significant empirical evaluation of what happens when those assumptions are violated or when other likely non-optimal conditions hold. Realistic conditions that might cause particular problems include measurement error and non-homogeneity of causal effect. Both of these are very likely to be present in the situations in which fairness is of concern (for example, crime, housing, loan approval, and admissions decisions). The paper would be significantly improved by experiments examining model accuracy and fairness under these conditions (and under errors in assumptions). The discussion of fairness and causal analysis omits at least one troubling aspect of real-world discrimination, which is the effect of continuing discrimination after fair decisions are made. For example, consider gender discrimination in business loans. If discriminatory actions of customers will reduce the probability of success for a female-owned business, then such businesses will have a higher probability of default, even if the initial loan decisions are made in an unbiased way. It would be interesting to note whether a causal approach would allow consideration of such factors and (if so) how they would be considered. Unmeasured selection bias would seem to be a serious threat to validity here, but is not considered in the discussion. Specifically, data collection bias can affect the observed dependencies (and, thus, the estimation of potential unmeasured confounders U and the strength of causal effects). Selection bias can occur due to selective data collection, retention, or reporting, when such selection is due to two or more variables. It is relatively easy to imagine selection bias arising in cases where discrimination and other forms of bias are present. The paper would be improved by discussing the potential effects of selection bias. A few minor issues: The paper (lines 164-166 and Figure 1) references Pearl's twin network, but the explanation of twin networks in caption of Figure 1 provides insufficient background for readers to understand what twin networks are and how they can be interpreted. Line 60 mentions "a wealth of work" two sentences after using "wealth" to refer to personal finances, and this may be confusing to readers.

Reviewer 2



Summary The authors compare several ideas of "fairness" in the ML literature and focus in on counter-factual fairness (to be defined in a second). The problem in general is, I observe some features X, some outcomes Y and some protected class of a person say A. I don't want my predictor to be "biased" against certain classes. How can I do this? I will focus on 3 key distinctions of fairness that the authors look at: 1) Unawareness (don't include A in model) 2) Individual fairness (make sure that if X is the same, then predictions are the same) 3) Causal Fairness: If we intervene and change A, does it change the prediction? The authors argue that causal fairness is the "right" definition of fairness. I find this very intuitive and I like the specifically causal way of looking at things. I think some parts of the paper are quite nice (I think the red car/policing example is actually the most illuminating of the whole thing, it's a shame it comes in so late in the paper) but others are quite confusing. Overall I think this is a "concept" paper rather than a particular technique. For this reason I am a weak accept on this paper. I am positive because I think it is important, but negative because a concept paper should be much clearer/more well polished. I now discuss ways I think the authors can improve the presentation of the paper without changing the actual content. Review 0) The paper can be made clearer by being more example driven The red car/crime examples are excellent, but they come in too late. They should come first, right after the definitions to show how one or another can fail. That example also explains quite well why we cannot condition on things downstream of A (Lemma 1). Similarly, the example of LSAT/GPA/FYA prediction in Figure 2 really explains everything. I would rather see the definitions + detailed explanation for the DAGS for the 2 examples + a simple linear version of the everything/the algorithm + the experiments + the experiment from the supplement on stops/criminality rather than the general formulation (put that in the supplement!). This would mean section 2.2 is not needed, one could write a simpler definition of Def 5. 1) It's hard to think about any guarantees of the fair learning algorithm. It wasn't clear to me what assumptions it requires for either a) some notion of asymptotic consistency of b) some notion of performing well (or failing). 2) It's not clear that the full algorithm is needed Looking at the Fair Prediction problem in section 5 (for Figure 2, level 2), why can't we just residualize race/sex out of the LSAT/FYA/GPA scores and then compute knowledge as the first principal component of the residuals? That seems to be quite close to what the authors are doing anyway and doesn't require any Bayesian modeling. I understand that the Bayesian mdoel provides posterior estimates etc... but conceptually, is there anything more to the model than that?

Reviewer 3



This paper addresses the question of whether a machine learning algorithm is "fair". Machine learning algorithms are being increasingly used to make decisions about people (e.g. hiring, policing, etc.). In some cases, making such decisions based on certain “protected” attributes may result in better predictions, but unfair decisions. For example, suppose that we wish to predict whether a particular person is likely to default on a loan and use that prediction to decide whether to accept or reject his/her loan application. If employers are discriminatory, then the race of the individual affects their employment, which in turn affects their ability to repay the loan (see Figure 1). As a result, predictions are likely to be more accurate if the race of the individual is used. However, it is unfair (and illegal) to reject a loan application on the basis of race. This paper proposes a definition of fairness for a predictor, which they call “counterfactual fairness”. This definition formalizes the intuition that a prediction is fair with respect to a protected attribute, say race, if that prediction would be the same had that person been born a different race. The authors discuss provide several compelling examples illustrating how this definition works and how it compares to prior conceptions of fairness. Next, the authors describe how to derive counterfactually fair predictors using causal assumptions of varying strength. Lastly, they apply their method to fairly predict success in law school. As expected, the counterfactually fair predictors perform worse than a "full" predictor that uses all of the available data. However, the full predictor would provide differential treatment to individuals had they been born a different race or sex. Also as expected, using stronger causal assumptions results in better predictions. Both counterfactually fair predictors also performed than a "fairness through unawareness" predictor that does not explicitly use any protected features for prediction. As machine learning takes a more prominent role in decision making related to people, the question of fairness becomes increasingly important. The proposed definition is compelling and, in my opinion, better captures human conceptions of fairness compared to prior, non-counterfactual definitions. Additionally, the paper is well written, easy to follow, and technically correct.